# CAPTURING MUSICAL STRUCTURE USING CONVOLUTIONAL RECURRENT LATENT VARIABLE MODEL

**Eunjeong Stella Koh**[*]**, Dustin Wright**[*] **& Shlomo Dubnov**
University of California, San Diego
* denotes equal contribution

## ABSTRACT

In this paper, we present a model for learning musical features and generating novel sequences of music. Our model, the Convolutional-Recurrent Variational Autoencoder (C-RVAE), captures short-term polyphonic sequential musical structure using a Convolutional Neural Network as a front-end. To generate sequential data, we apply the recurrent latent variational model, which uses an encoder-decoder architecture with latent probabilistic connections to capture the hidden structure of music. Using the sequence-to-sequence model, our generative model can exploit samples from a prior distribution and generate a longer sequence of music.

## 1 INTRODUCTION

Previous studies (Chen & Miikkulainen 2001; Waite et al.; Boulanger-Lewandowski et al. 2012) on Recurrent Neural Network (RNN) for music generation face two major challenges related to: 1) understanding the higher level semantics of musical structure, which is critical to music composition, and 2) simple but repetitive patterns in the musical output (Bretan et al. 2016). The musical surface is difficult to represent when dealing with multiple instruments within a piece (polyphony). To address this, many music generation models rely on pre-processing of the musical features prior to training. In this study, the difficulty in representing polyphony is handled using a Convolutional Neural Network (CNN) acting on the symbolic representation of the input music. CNNs have been utilized for music generation in previous work starting from audio-domain music data (i.e., wav) (Oord et al. 2016), and recent studies introduce a CNN with symbolic-domain data, leading to innovations in music generation with complex melodies. C-RNN-GAN (Mogren 2016), MidiNet (Yang et al. 2017), and MuseGAN (Dong et al. 2017) developed different models, which encourage researchers to use CNN for capturing musical structure. Several approaches have been studied for: 1) covering multi-channel MIDIs by CNN layers, 2) setting several generative models for multi-track data generation, and 3) processing longer sequences of data. As an extension of these advancements, we explore the Convolutional-Recurrent Variational Autoencoder (C-RVAE), which is an effective method of learning useful musical features that we use for polyphonic music generation. In the studies by Hennig et al. (2017); Roberts et al., the variational autoencoder (VAE) has been shown to be useful for musical creation. In this same vein, the Variational Recurrent Neural Network, introduced in Fabius & van Amersfoort 2014; Chung et al. 2015, was shown to perform well on generating sequential outputs by integrating latent random variables in a recurrent neural network. For the latent variable structure, the model utilizes encoded data in latent space in each step, and the studies argue that these recurrent steps can make it possible to be flexible on the generation of more diverse styles of music while incorporating features from data in a concrete way. With these possibilities in mind, we propose that our model can extract musical structure using the VAE structure in a recurrent network combined with the CNN for learning a representation of symbolic domain music.

## 2 METHOD

### 2.1 FEATURE LEARNING WITH CONVOLUTIONAL NEURAL NETWORK

We adopt a CNN in order to learn a better representation of polyphonic music by treating the input as a 2D binary feature map. This is predicated on the notion that the arrangement of notes in a musical

piece contains salient spatial relationships when visualized in a form such as a piano-roll, and thus are conducive to being modeled by a CNN. The input MIDI is first preprocessed into a piano-roll, with the beat resolution set to $8^{\text{th}}$ notes. This gives us a feature map representation $\mathbf{x}^{(t)} \in \{0, 1\}^{n \times r \times 1}$ at time step $t$, where $n$ is a number of time steps in a frame and $r$ is the note range. The piano-roll is then processed with two Convolutional layers separated by max-pooling layers and a final flattening layer. The output of this network is the latent feature $\mathbf{m}_l^{(t)} \in \mathbb{R}^k$ at time step $t$ (See panel A).

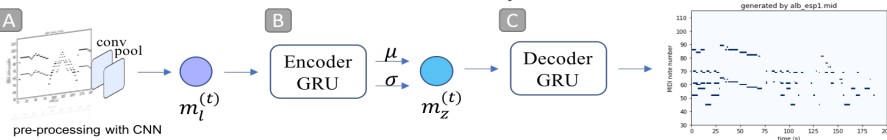

Figure 1: Network Structure of the Convolutional-Recurrent Variational Autoencoder (C-RVAE)

## 2.2 ENCODER & DECODER NETWORK

We combine a recurrent architecture with a Variational Autoencoder (VAE), which includes an RNN encoder and RNN decoder (See panel B and C). Here are some properties of the VAE structure:

- The VAE consists of a decoding network with parameters $\theta$, which estimates the posterior distribution $p_\theta(\mathbf{x}|\mathbf{z})$, where $\mathbf{x}$ is the sample being estimated and $\mathbf{z}$ is an unobserved continuous random variable.
- The true posterior distribution $p_\theta(\mathbf{z}|\mathbf{x}) = p_\theta(\mathbf{x}|\mathbf{z})p_\theta(\mathbf{z})/p_\theta(\mathbf{x})$ is intractable, so an encoding network $q$ with parameters $\phi$ is used to estimate the posterior as $q_\phi(\mathbf{z}|\mathbf{x})$.

The encoder RNN takes the latent feature $\mathbf{m}_l^{(t)}$ at each time step and produces a final hidden state $\mathbf{h}_q^{(T)} \in \mathbb{R}^e$ for a sequence of $T$ MIDI frames ($\mathbf{h}_q^{(T)} = f_{\text{RNN}}(\mathbf{m}_l^{(1)}, ..., \mathbf{m}_l^{(T)})$). The hidden state is then subject to two linear transformations to determine the mean & standard deviation ($\boldsymbol{\mu} = \mathbf{W}_\mu \mathbf{h}_q^{(T)} + \mathbf{b}_\mu$ & $\boldsymbol{\sigma} = \mathbf{W}_\sigma \mathbf{h}_q^{(T)} + \mathbf{b}_\sigma$) of the noise distribution which is given as $\log q_\phi(\mathbf{z}|\mathbf{x}) = \log \mathcal{N}(\mathbf{z}; \boldsymbol{\mu}, \boldsymbol{\sigma}^2 \mathbf{I})$. In this formulations, $\mathbf{W}_\mu, \mathbf{W}_\sigma \in \mathbb{R}^{z \times e}$ and $\mathbf{b}_\mu, \mathbf{b}_\sigma \in \mathbb{R}^z$. noise is then generated as $\mathbf{z} = \boldsymbol{\mu} + \boldsymbol{\sigma} \odot \boldsymbol{\epsilon}, \boldsymbol{\epsilon} = \mathcal{N}(0, \mathbf{I})$.

Since we are modeling sequential data, the decoder network is trained to predict $p_\theta(\mathbf{x}^{(t)}|\mathbf{x}^{(1:t-1)}, \mathbf{z})$. The RNN takes in the generated noise $\mathbf{z}$ at the first time step. At each subsequent time step, the latent feature $\mathbf{m}_l^{(t)}$ for an input sample $\mathbf{x}^{(t)}$ is linearly transformed into the same dimensionality as the noise source and then passed into the RNN, $\mathbf{m}_z^{(t)} = \mathbf{W}_z \mathbf{m}_l^{(t)} + \mathbf{b}_z$ ($\mathbf{W}_z \in \mathbb{R}^{z \times k}, \mathbf{b}_z \in \mathbb{R}^z$). The RNN produces a hidden state $\mathbf{h}_p^{(t)}$ at each time step, which is passed through a logistic layer estimating $p_\theta(\mathbf{x}^{(t)}|\mathbf{x}^{(1:t-1)}, \mathbf{z})$.

During training, the model is presented with samples of the input, which are encoded by $q$ to produce the mean and standard deviation for the noise source. A noise sample is then generated and passed through the decoding network, which calculates the posterior probability $p$ to determine the sample generated by the network. The network is trained to reproduce the input sample from the noise source, so the second term on the right-hand side of $\mathcal{L}(x; \theta, \phi) \simeq \frac{1}{2} \sum_j (1 + \log(\boldsymbol{\sigma}_j^2) - \boldsymbol{\mu}_j^2 - \boldsymbol{\sigma}_j^2) + \frac{1}{L} \sum_l \log p_\theta(\mathbf{x}|\mathbf{z}^{(l)})$ can be either mean squared error in the case of a continuous random variable or cross entropy for discrete random variables. At test time, random samples are generated by the noise source, which is used by the decoder network to produce novel sequences. Finally, we can calculate $\mathbf{h}_p^{(t)} = f_{\text{RNN}}(\mathbf{z}, \mathbf{m}_z^{(1)}, ..., \mathbf{m}_z^{(t)})$ and $\tilde{\mathbf{x}}^{(t)} = \sigma(\mathbf{W}_p \mathbf{h}_p^{(t)} + \mathbf{b}_p)$, where $\sigma(\cdot)$ is the logistic sigmoid function, $\mathbf{h}_p^{(t)} \in \mathbb{R}^d$, and $\mathbf{W}_p \in \mathbb{R}^{nr \times d}$. This effectively yields a binary feature map of the same dimensionality as the input, which is used to predict a piano-roll based on the input at the previous time steps and the noise. We use the Gated Recurrent Unit (GRU) (Cho et al. 2014) for both the encoder and decoder RNN. The model generates a track of music bar by bar, with a possibly polyphonic structure captured among several bars.

## 3 EXPERIMENTS

In our implementation, we use 256 hidden units for the encoder and 512 hidden units for the decoder. Moreover, we use Adam optimizer by Kingma & Welling (2013) with a learning rate of $1 \cdot 10^{-4}$

over 2k epochs. The training MIDI data[1] were processed into 512 individual time steps, utilizing the encoded tempo to determine the time resolution such that each time step was one $8^{th}$ note. At each time step of training, we set a frame of music to be half of a bar and train on 8 bars at a time such that we are reconstructing 16 frames in each training step.

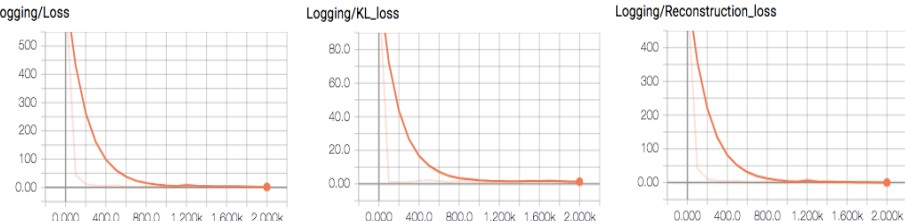

Figure 2: Approximating our distribution. Loss function, KL-loss function, and Reconstruction loss function were computed during training.

For proof of concept, a composer participated in the listening test of our generated musical output. One of the most difficult challenges in music composition by neural networks is to verify the quality of the music itself. To verify our model and results, we analyzed the musical structure with the composer. We randomly selected 10 generation results of the C-RVAE model and composers were asked to describe the characteristics of each generated sequence. *"The computer almost catches the primary structure of melodic skeleton. And the original chord progression is being reduced to 3 chords, which forms a structured music."*, *"The ending chord is very satisfied since it isn't on a clear and strong musical cadence. If the bass note D could be replaced by the G, it might be more persuasive."* The C-RVAE model can compose music more dynamically while including the original theme. Our sample results are also posted on soundcloud[2].

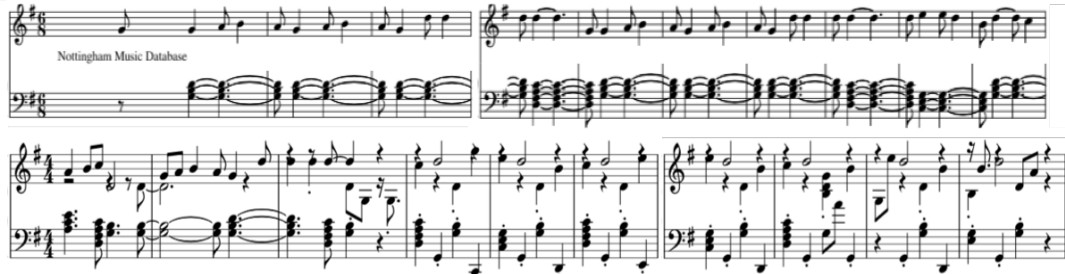

Figure 3: Comparison of music used as input and generated result. Top: The first 10 bars of the training input data / Bottom: The first 10 bars of the generated musical output

## 4 DISCUSSION

In this study, we are able to show initial proof that the C-RVAE model applied to MIDI sequence representations can capture the structure of and create polyphonic music. We use an encoder-decoder architecture with latent probabilistic connections to understand the hidden structure of music, such as harmony. While it has been challenged, controlling the latent space still needs to be improved for the musical output. In this model, random sampling and data interpolation can compose music more dynamically while including the original theme (See Figure 3). Most previous studies for music generation use so called one-to-many RNN, where single musical units (such as a single note or one bar of music) has been used to predict the next unit in a recurrent manner. At the application stage of our method, we introduce the model to emulate a specific song from a video game and generate background music similar in style to those examples. In addition, some musical applications need to work with fewer samples in order to generate a specific musical result, and our C-RVAE model is meant to allow that.

---

[1] four of VGMusic(vgmusic.com), two of Nottingham data, three of Piano-midi.de, one of TheoryTab data
[2] https://soundcloud.com/user-431911640/sets

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
