# OpenReview forum: "Capturing Musical Structure Using Convolutional Recurrent Latent Variable Model"
_ICLR.cc/2018/Workshop — Reject_

### Official Review · AnonReviewer1 · 2018-03-08
**Did convolution improve the model?**

**Rating:** 5
**Confidence:** 3

**Review:**

This paper proposes to extend the Variational Recurrent Autoencoder model from Amersfoort and Fabius (2014) by using convolutional layers in addition to the recurrent inference network and apply this method to MIDI sequences to demonstrate the capability of the model.
The model is trained on 10 training examples and the audio samples in the soundcloud link only demonstrates the reconstruction ability of the model. This brings up an unaddressed question on whether the model overfits or not. Is it able to generate novel music in a consistent fashion?
Moreover, it is hard from the paper to tell whether there was any improvement resulting from using convolutions. Are the numbers better? Are the samples better (e.g. according to the composers you asked)?

---

### Official Review · AnonReviewer3 · 2018-03-09
**Nice application but limited evaluation and not that novel.**

**Rating:** 4
**Confidence:** 5

**Review:**

The paper describes a VAE-RNN based generative model for polyphonic midi music. A convolutional network is used to learn input representations and the sequences generated by the model are evaluated by composers.

Despite the limited space, the abstract was relatively easy to follow and clear. This is impressive considering the relative complexity of the described model.

As the text points out, VAE-based recurrent models have been  around since 2014 and the approach is not new except perhaps for the use of convolutional input transformations. However, such convolutional transformations have been combined with RNN models before as well (again, the authors acknowledge this, so this is no complaint about credit assignment). This lack of novelty could be offset by impressive results or scaling the models up to challenging data sets but the data sets used in the paper are very small and no quantitative results are reported. The composer quotations are hard to interpret because the paper doesn't specify how many composers were involved and there is no baseline system to compare with. The samples on soundcloud sound nice, but it is unfortunate that it is not clear whether they are generated from noise or reconstructed from input sequences.

Pros: Clear presentation, good sounding samples

Cons: Lack of novelty, limited evaluation, small data set

---

### Official Review · AnonReviewer2 · 2018-03-09
**Not good enough**

**Rating:** 3
**Confidence:** 4

**Review:**

The authors of this paper claim that using convolutional variant of recurrent variational auto-encoders (C-RVAEs) can learn musical structures. I have read the paper carefully and also listened to the samples provided by the authors.

Here are some points that I am concerned about this paper.
1. The authors do not compare their method with any other baseline models. It is important to compare with other baseline models, otherwise we don't know if it is this particular model that works well on the task or any kind of models can work well with the same experimental setting.
2. The authors are using the paper space for no good reason. For instance, Figure 2 does not give any insightful information.
3.  Polyphonic music datasets are known to have very small number of examples. It is hard to judge how over-fitting was handled in the paper.

---

### Decision · Program_Chairs · 2018-03-20
**ICLR 2018 Workshop Acceptance Decision**

**Decision:**

Reject

**Comment:**

Based on the reviews, this paper has not been accepted for presentation at the ICLR workshop. However, the conversation and updates can continue to appear here on OpenReview.